# Evaluating the Role of Hepatobiliary Phase of Gadoxetic Acid-Enhanced Magnetic Resonance Imaging in Predicting Treatment Impact of Lenvatinib and Atezolizumab plus Bevacizumab on Unresectable Hepatocellular Carcinoma

**DOI:** 10.3390/cancers14030827

**Published:** 2022-02-06

**Authors:** Ryu Sasaki, Kazuyoshi Nagata, Masanori Fukushima, Masafumi Haraguchi, Satoshi Miuma, Hisamitsu Miyaaki, Akihiko Soyama, Masaaki Hidaka, Susumu Eguchi, Masaya Shigeno, Mio Yamashima, Shinobu Yamamichi, Tatsuki Ichikawa, Yuki Kugiyama, Hiroshi Yatsuhashi, Kazuhiko Nakao

**Affiliations:** 1Department of Gastroenterology and Hepatology, Nagasaki University Graduate School of Biomedical Sciences, 1-7-1 Sakamoto, Nagasaki City 852-8501, Nagasaki, Japan; kazuyoshi.ngt@nagasaki-u.ac.jp (K.N.); ma-fukushima@nagasaki-u.ac.jp (M.F.); mharaguchi@nagasaki-u.ac.jp (M.H.); miuma1002@gmail.com (S.M.); miyaaki-hi@umin.ac.jp (H.M.); kazuhiko@nagasaki-u.ac.jp (K.N.); 2Department of Surgery, Nagasaki University Graduate School of Biomedical Sciences, 1-7-1 Sakamoto, Nagasaki City 852-8501, Nagasaki, Japan; soyama@nagasaki-u.ac.jp (A.S.); mahidaka@nagasaki-u.ac.jp (M.H.); sueguchi@nagasaki-u.ac.jp (S.E.); 3Department Gastroenterology and Hepatology, Japanese Red Cross, Nagasaki Genbaku Hospital, 3-15 Mori-machi, Nagasaki City 852-8511, Nagasaki, Japan; mshi1010@nagasaki-med.jrc.or.jp; 4Department Gastroenterology and Hepatology, Nagasaki Harbor Medical Center, 6-39 Shinchi-machi, Nagasaki City 850-8798, Nagasaki, Japan; mamio5612@gmail.com (M.Y.); bwspn147@yahoo.co.jp (S.Y.); ichikawa@nagasaki-u.ac.jp (T.I.); 5Clinical Research Center, National Hospital Organization Nagasaki Medical Center, Department of Hepatology, 2-1001-1 Kubara, Omura City 856-8562, Nagasaki, Japan; kugiyama.yuki.kn@mail.hosp.go.jp (Y.K.); yatsuhashi.hiroshi.wk@mail.hosp.go.jp (H.Y.)

**Keywords:** atezolizumab, bevacizumab, biomarkers, hepatocellular carcinoma, catenins, lenvatinib

## Abstract

**Simple Summary:**

Atezolizumab plus bevacizumab therapy has high response rates in patients with unresectable hepatocellular carcinoma (HCC). The hepatobiliary phase of gadoxetic acid-enhanced magnetic resonance imaging (EOB-MRI) has been reported to be useful as an imaging biomarker for detecting β-catenin mutations. We evaluated whether pretreatment in the hepatobiliary phase of EOB-MRI could predict the therapeutic effect of lenvatinib (*n* = 33) and atezolizumab plus bevacizumab (*n* = 35). The visual assessment and relative enhancement ratio (RER) of the largest HCC lesions were evaluated using the hepatobiliary phase of EOB-MRI. In the lenvatinib group, progression-free survival (PFS) was not differently stratified using EOB-MRI. In the atezolizumab plus bevacizumab group, the heterogeneous type had significantly shorter PFS than the homogenous type, and the hyperintensity (RER ≥ 0.9) type had significantly shorter PFS than the hypointensity type. Hence, the hepatobiliary phase of EOB-MRI was useful for predicting the therapeutic effect of atezolizumab plus bevacizumab therapy on unresectable HCC.

**Abstract:**

Background: Atezolizumab plus bevacizumab therapy has high response rates in patients with unresectable hepatocellular carcinoma (HCC). The hepatobiliary phase of gadoxetic acid-enhanced magnetic resonance imaging (EOB-MRI) has been reported to be useful as an imaging biomarker for detecting β-catenin mutations. We evaluated whether the pretreatment of the hepatobiliary phase of EOB-MRI could predict the therapeutic effect of lenvatinib and atezolizumab plus bevacizumab. Methods: This study included 68 patients (lenvatinib group (*n* = 33) and atezolizumab plus bevacizumab group (*n* = 35)). The visual assessment and relative enhancement ratio (RER) of the largest HCC lesions were evaluated using the hepatobiliary phase of EOB-MRI. Results: The hyperintensity type (RER ≥ 0.9) was 18.2% in the lenvatinib group and 20.0% in the atezolizumab plus bevacizumab group. In the lenvatinib group, progression-free survival (PFS) was not different between the heterogeneous and homogenous types (*p* = 0.688) or between the hyperintensity and hypointensity types (*p* = 0.757). In the atezolizumab plus bevacizumab group, the heterogeneous type had significantly shorter PFS than the homogenous type (*p* = 0.007), and the hyperintensity type had significantly shorter PFS than the hypointensity type (*p* = 0.012). Conclusions: The hepatobiliary phase of EOB-MRI was useful for predicting the therapeutic effect of atezolizumab plus bevacizumab therapy on unresectable HCC.

## 1. Introduction

Systemic therapy for unresectable hepatocellular carcinoma (HCC) has made great strides in recent years. In addition to molecular-targeted agents, immune checkpoint inhibitors (ICIs) have become available, expanding treatment options [1,2]. Atezolizumab plus bevacizumab therapy was the first regimen to show superiority to sorafenib and is currently the first-line treatment of unresectable HCC [2,3]. High response rates could be achieved with atezolizumab plus bevacizumab therapy. However, no established biomarkers have been found to predict this response.

In other types of carcinomas, β-catenin mutations suppress antitumor immunity [4]. Recently, it has been reported that ICI monotherapy had poor therapeutic effects in patients with HCC carrying Wnt/β-catenin mutations [5,6]. In addition, the immunological tumor microenvironment is important for improving the therapeutic effect of ICI therapy [7]. Therefore, histopathological evaluation is required for detecting β-catenin mutations. Moreover, in recent years, the hepatobiliary phase of gadoxetic acid-enhanced magnetic resonance imaging (EOB-MRI) has been reported as a useful imaging biomarker for detecting β-catenin mutations [8]. The relative enhancement ratio (RER) and visual assessment have been used for assessing signal intensity in the EOB-MRI hepatobiliary phase. It has been reported that typical HCCs have low signal intensity (hypointensity type) on assessment based on the RER in the EOB-MRI hepatobiliary phase. Hyperintensity-type HCCs have a high frequency of β-catenin mutations [8]. However, it has been reported by visual assessment that typical HCCs have low homogenous signal intensity in the EOB-MRI hepatobiliary phase. Lesions with heterogeneous signal intensity types had poor prognoses [9,10]. Furthermore, EOB-MRI has been reported as a promising imaging-based biomarker for predicting unfavorable responses to ICI monotherapy for unresectable HCC [11]. However, there have been insufficient reports on the relationship between signal intensity on the EOB-MRI hepatobiliary phase and combination therapy with ICI and antivascular endothelial growth factor (VEGF) inhibitors.

In this study, we evaluated whether an assessment using the EOB-MRI hepatobiliary phase could predict the therapeutic effects of lenvatinib and atezolizumab plus bevacizumab therapy.

## 2. Materials and Methods

### 2.1. Patients

From March 2018 to November 2021, there were 76 (except duplicate cases) patients with unresectable HCC who underwent EOB-MRI 3 months prior to treatment initiation. In addition, they were administered lenvatinib or atezolizumab plus bevacizumab therapy at Nagasaki University Hospital and its related facilities. This study included 68 patients allocated to the lenvatinib group (*n* = 33) and atezolizumab plus bevacizumab group (*n* = 35). Patients with lenvatinib administration period of less than 3 weeks (*n* = 5) and in whom treatment response could not be determined (*n* = 3) were excluded from the study.

### 2.2. Treatment Protocol, Evaluation Criteria for Response, and Followup of HCC

Lenvatinib was orally administered to the patients weighing less than 60 kg or with Child–Pugh grade B and over 60 kg, at doses of 8 mg and 12 mg, respectively. Intravenous treatment with 1200 mg atezolizumab plus 15 mg/kg of body weight bevacizumab was administered every 3 weeks. Treatment was discontinued when unacceptable adverse events or clinical tumor progression was observed. Treatment response was evaluated using contrast-enhanced computed tomography or magnetic resonance imaging with the Response Evaluation Criteria in Solid Tumors (RECIST) [12] and modified RECIST (mRECIST) [13] every 8–12 weeks. The best response was adopted as therapeutic effect.

### 2.3. Image Analysis

HCC was evaluated by the EOB-MRI hepatobiliary phase using 1.5 T Achieva (Philips, Best, The Netherlands) and 3.0 T Magnetom Skyra (Siemens Healthineers, Erlangen, Germany) at facility A. In addition, 3.0 T Magnetom Skyra was used at facility B, C, and D, respectively. According to previously described methods, EOB-MRI images of the largest HCC lesions were independently evaluated by two hepatologists with more than 10 years of experience in HCC treatment [8]. Each lesion was divided into two groups, a homogenous group and a heterogeneous group, based on visual assessment. In addition, signal intensity (SI) of tumor lesion and nontumor liver tissue was measured by defining a region of interest and calculated according to previously reported methods [8,14]. Hyperintensity was defined as an RER of ≥0.9, according to a previous report [8]. Moreover, the RER was calculated as follows: (nodule SI/parenchyma SI on hepatobiliary phase images)/(nodule SI/parenchyma SI on precontract images).

Concordance rate between the two observers performing visual assessments was evaluated using weighted κ statistics. After evaluating visual assessments in a consensus fashion, we performed data analysis. Concordance rate between the two observers performing the RER was evaluated using the intraclass correlation coefficient (ICC). Furthermore, the RER was calculated by adopting an average value.

### 2.4. Ethical Considerations

This study was based on the 1964 Declaration of Helsinki and the ethical standards (institutional and national) of the committee responsible for human experimentation. Informed consent was obtained from all patients in advance. The study protocol was approved (16 April 2019) by the Ethical Committee of Nagasaki University Hospital (approval number: 19041523-4).

### 2.5. Statistical Analysis

To estimate progression-free survival (PFS) rate after drug administration, we used the Kaplan–Meier method and log-rank test. Weighted κ and ICC values were qualified as: <0.20 (poor agreement), 0.20–0.39 (fair agreement), 0.40–0.59 (moderate agreement), 0.60–0.79 (substantial agreement), and >0.80 (excellent agreement). Receiver operating characteristic (ROC) curves were assessed to select the optimal cut-off values for the RER. Statistical significance was set at *p* < 0.05. Data analysis was performed using the SPSS ver. 22.0 (SPSS, Chicago, IL, USA).

## 3. Results

### 3.1. Patient Characteristics

The baseline characteristics of the 68 patients enrolled in this study are summarized in Table 1. The median observation periods after treatment with lenvatinib and atezolizumab plus bevacizumab therapy were 16.0 months and 5.3 months, respectively.

### 3.2. HCC Image Analysis by EOB-MRI Hepatobiliary Phase

Regarding visual assessment (heterogeneous vs. homogenous), the weighted κ value for agreement between the two observers was 0.793 (substantial agreement). Regarding the RER, the ICC value for agreement between the two observers was 0.819 (excellent agreement). Table 2 shows the evaluation of the EOB-MRI hepatobiliary phase and the largest lesions of HCC in the lenvatinib and atezolizumab plus bevacizumab groups. The heterogeneous type by visual assessment was 36.4% in the lenvatinib group and 45.7% in the atezolizumab plus bevacizumab group. The hyperintensity type with an RER of ≥0.9 was 18.2% in the lenvatinib group and 20.0% in the atezolizumab plus bevacizumab group.

### 3.3. Relationship between PFS and Assessment of EOB-MRI Hepatobiliary Phase

We assessed PFS stratified by visual assessment using EOB-MRI. In the lenvatinib group, there was no difference in PFS between the heterogeneous and homogenous types (*p* = 0.688, Figure 1a). In the atezolizumab plus bevacizumab group, the heterogeneous type had a significantly shorter PFS than the homogenous type (*p* = 0.007, Figure 1b). Furthermore, we assessed PFS stratified by the RER. In the lenvatinib group, there was no difference in PFS between the hyperintensity and hypointensity types (*p* = 0.757, Figure 2a). In the atezolizumab plus bevacizumab group, the hyperintensity type had significantly shorter PFS than the hypointensity type (*p* = 0.012, Figure 2b).

### 3.4. Response Rate Stratified by Assessment of EOB-MRI Hepatobiliary Phase

Table 3 shows the response rates stratified by assessment of the EOB-MRI hepatobiliary phase in the lenvatinib and atezolizumab plus bevacizumab groups. The disease control rate in the lenvatinib group was 66.7% in the heterogeneous type and 62.5% in the hyperintensity type. The disease control rate in the atezolizumab plus bevacizumab group was 66.7% in the heterogeneous type and 42.9% in the hyperintensity type.

### 3.5. Relationship between Visual Assessment and RER in EOB-MRI Hepatobiliary Phase

Table 4 shows the relationship between visual assessment and RER in the EOB-MRI hepatobiliary phase in all 68 cases. Visual assessment of the EOB-MRI hepatobiliary phase showed that 97.5% of the homogenous type was hypointense. On the other hand, 57.1% of the heterogenous type was hypointense, and 42.9% of the heterogenous type was hyperintense.

### 3.6. Prediction Ability of Response by RER

The predictive ability of response (disease control) was evaluated using the receiver operating characteristic (ROC) curve for the RER. The area under the ROC curve was 0.627, and the optimal cut-off value of the RER according to the ROC curve was 0.88.

## 4. Discussion

It has been shown that ICIs are effective in many carcinomas, including HCC, and depending on cancer type, biomarkers can be used to predict their therapeutic effects [15]. However, there is no effective biomarker for HCC that can predict the therapeutic effect of atezolizumab plus bevacizumab therapy, which is an unmet need. In the present study, we evaluated whether pretreatment EOB-MRI could be useful for predicting response to lenvatinib and atezolizumab plus bevacizumab therapy for HCC.

The first main finding of our study was that the heterogeneous type assessed by the EOB-MRI hepatobiliary phase was approximately 40%, and the hyperintensity type was approximately 20% in large size lesions in patients with unresectable HCC (Table 2). These results are consistent with those of previous studies [9,16]. Furthermore, both the visual assessment and RER showed high concordance rates, evaluated individually by the two observers.

Except for one case, all cases of the homogenous type identified by visual assessment (97.5%) were hyperintense (RER ≥ 0.9). However, among the heterogeneous types, the hyperintensity type (RER ≥ 0.9) was 42.9% (Table 4). There was a high probability that the RER may be less than 0.9 if a target lesion was homogenous by visual assessment. Although visual assessment is easy to evaluate, it is not objective. However, the concordance rate among the interobservers was high, and it was considered useful in clinical practice.

Of the hyperintensity type (RER ≥ 0.9), 92.3% were heterogeneous, although the number of cases of hyperintensity type was small, and it was difficult to examine in detail cases of a high RER. In addition, the RER has a problem with cut-off values, and there are reports of 0.9 and 1.0 in previous studies [8,16], with no fixed value. In this study, the cut-off value that could predict the disease control of the RER was 0.88. Therefore, it might be appropriate to use the previously reported cut-off value of 0.9. Although the RER is more objective than visual assessment, it is slightly more complicated in clinical practice.

The second main finding of our study was that assessment by the EOB-MRI hepatobiliary phase did not affect PFS in the lenvatinib group. Kubo et al. reported that lenvatinib response did not change on the EOB-MRI hepatobiliary phase at even high intensity [16], which also supports our results. One of the reasons for this is that lenvatinib does not reduce the therapeutic effect on HCC in patients with β-catenin mutations [17]. There is a correlation between Wnt/β-catenin signaling and fibroblast growth factor receptor (FGFR) 4 expression. Furthermore, lenvatinib has been reported to have a high response to high FGFR4 expression [18]. It has also been reported that the therapeutic effect of lenvatinib did not change with tumor differentiation [19], and it can be expected to be effective in various types of tumors.

The third main finding of our study was that assessment by the EOB-MRI hepatobiliary phase affected PFS in the atezolizumab plus bevacizumab group (Figure 1 and Figure 2). We also examined liver function and the neutrophil-to-lymphocyte ratio (NLR) that have been reported to be associated with PFS for atezolizumab plus bevacizumab therapy, but found no statistical differences [20,21]. One of the reasons for predicting therapeutic effects is that the heterogeneous and hyperintense types were present in a certain number of patients with HCC carrying β-catenin mutation. However, the prediction rate of β-catenin mutation assessed by EOB-MRI was approximately 80% [8]; this finding alone remains unexplained, and other factors are considered. Heterogeneous and hyperintense types are large in size and may reflect the degree of differentiation within HCC and the nonuniformity of molecular biological characteristics. In addition, it is important to note that there were cases in which disease control was achieved by the atezolizumab plus bevacizumab therapy even with heterogeneous or hyperintense type (Table 3). These results do not necessarily mean that atezolizumab plus bevacizumab therapy is not effective due to the presence of heterogeneous or hyperintense types. This is expected to be related to the therapeutic effect of bevacizumab, an antibody that targets VEGF A [22]. Bevacizumab promotes the maturation of myeloid cells and dendritic cells, normalizes tumor blood vessels, and augments intratumoral T-cell infiltration [23,24,25]. The role of bevacizumab is thought to transform suppressive immune microenvironments into responsive ones. For these reasons, even in the presence of the WNT/β-catenin mutation, there are cases in which the therapeutic effect of atezolizumab plus bevacizumab therapy could be expected [26]. Regarding atezolizumab plus bevacizumab therapy for heterogeneous or hyperintense types HCC, it may be necessary to make an early judgment of tumor response in clinical practice. Similarly, we suggest that more clinical cases need to be evaluated to determine whether β-catenin mutations affect the therapeutic effect of atezolizumab and bevacizumab therapy for unresectable HCC.

One of the limitations of the present study is its retrospective nature. A future prospective analysis will be needed to validate the efficacy of the EOB-MRI hepatobiliary phase as a predictor of the therapeutic effect of atezolizumab plus bevacizumab therapy. Another limitation is that we analyzed data of a relatively small number of cases and performed a short-term analysis. The two observers who assessed the images were affiliated with our facility. This study was a multicenter analysis, and MRI models and settings are different at each facility. A histological evaluation was not performed. Therefore, it might not be possible to determine the extent to which β-catenin mutation could have affected this result. Therefore, to address these limitations, it is necessary to increase the number of cases in the future by extending the facility and observation period in addition to independent image evaluation.

Regardless of these limitations, this is the first study to report the relationship between assessment by the EOB-MRI hepatobiliary phase and predicting therapeutic effects of atezolizumab plus bevacizumab therapy in patients with unresectable HCC.

## 5. Conclusions

In conclusion, the EOB-MRI hepatobiliary phase was useful for predicting the therapeutic effect of atezolizumab plus bevacizumab therapy on unresectable HCC.

## Figures and Tables

**Figure 1 cancers-14-00827-f001:**
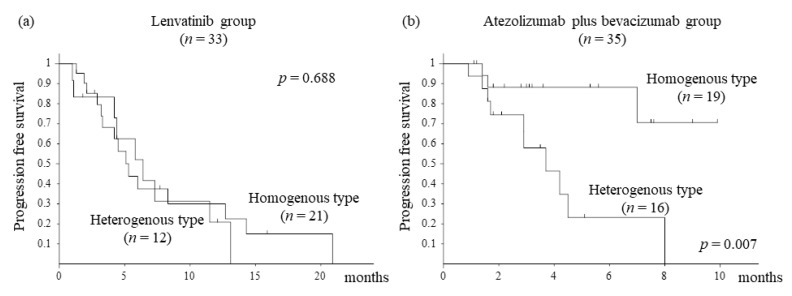
Progression-free survival stratified by EOB-MRI visual assessment: (**a**) Kaplan–Meier curve of progression-free survival (PFS) in the lenvatinib group (*n* = 33). The median PFS with homogenous and heterogeneous types is 5.1 months and 6.4 months, respectively. There is no significant difference in PFS between the homogenous and heterogeneous types (*p* = 0.688, log-rank test). (**b**) Kaplan–Meier curve of PFS in the atezolizumab plus bevacizumab group (*n* = 35). The median PFS is not attained with homogenous type and 6.4 months in heterogeneous type. Moreover, PFS is significantly better in the homogenous type than in the heterogeneous type (*p* = 0.007, log-rank test).

**Figure 2 cancers-14-00827-f002:**
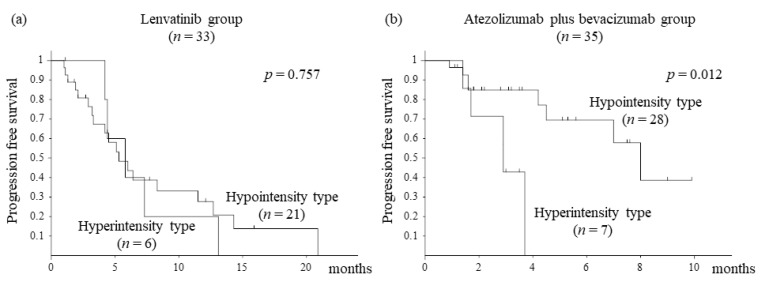
Progression-free survival (PFS) stratified by relative enhancement ratio: (**a**) Kaplan–Meier curve of PFS in the lenvatinib group (*n* = 33). The median PFS with hypointensity and hyperintensity types is 5.3 months and 6.4 months, respectively. There is no significant difference in PFS between the hypointensity and hyperintensity types (*p* = 0.757, log-rank test). (**b**) Kaplan–Meier curve of PFS in the atezolizumab plus bevacizumab group (*n* = 35). The median PFS with hypointensity type and hyperintensity type is 8.0 months and 2.9 months, respectively. Additionally, PFS was significantly better in the hypointensity type than in the hyperintensity type (*p* = 0.012, log-rank test).

**Table 1 cancers-14-00827-t001:** Characteristics of the patients enrolled in the present study.

Variable		Lenvatinib (*n* = 33)	Atezolizumab plus Bevacizumab (*n* = 35)
Age	Year	75.0 (51–84)	69.0 (48–88)
Sex	male/female	26/7	27/8
BMI	kg/m^2^	22.70 (16.5–35.9)	21.80 (16.9–28.6)
Performance status	0/1/2	26/7/0	23/10/2
Child–Pugh score	A/B	31/2	30/5
Macroscopic PV invasion	Vp3 or 4	5 (15.2%)	5 (14.2%)
Extrahepatic spread	+	6 (18.2%)	11 (31.4%)
BCLC stage	B/C	18/15	19/16
Etiology	HBV/HCV/NBNC	7/9/17	6/5/24
Platelet count	×10^4^/μL	15.70 (4.4–31.4)	13.70 (6.7–40.5)
T.bil	mg/dL	0.80 (0.3–2.2)	0.90 (0.3–2.0)
Albumin	g/dL	3.70 (2.7–4.7)	3.70 (2.2–5.4)
ALT	IU/mL	24.0 (7–137)	29.0 (13–87)
AFP	ng/mL	111.0 (2–89,533)	12.5 (2–48,400)
DCP	mAU/mL	539.0 (13–75,000)	1115.0 (14–75,000)
NLR	Ratio	2.60 (1.2–9.2)	3.10 (0.9–9.3)
Treatment period	Month	4.90 (1.0–38.0)	3.00 (0.7–9.9)
Period until dose reduction	Month	1.40 (0.2–14.4)	2.30 (0.7–9.9)

Data are presented as medians with ranges or numbers with percentages. BMI, body mass index; PV, portal vein; BCLC, Barcelona Clinic liver cancer; HBV, hepatitis B virus; HCV, hepatitis C virus; NBNC, non B non C; T.bil, total bilirubin; ALT, alanine aminotransferase; AFP, alpha fetoprotein; DCP, Des-gamma-carboxy prothrombin; NLR, neutrophil-to-lymphocyte ratio.

**Table 2 cancers-14-00827-t002:** Image analysis by EOB-MRI hepatobiliary phase.

Factors		Lenvatinib (*n* = 33)	Atezolizumab plus Bevacizumab (*n* = 35)
Tumor size	cm	3.20 (1.1–12.8)	4.00 (1.0–19.0)
Visual assessment	Homogenous type	21 (63.6%)	19 (54.3%)
	Heterogenous type	12 (36.4%)	16 (45.7%)
RER	Value	0.79 (0.56–1.38)	0.76 (0.50–1.18)
	Hypointensity type (RER <0.9)	27 (81.8%)	28 (80.0%)
	Hyperintensity type (RER ≥0.9)	6 (18.2%)	7 (20.0%)

Data are presented as medians with ranges or numbers with percentages. RER, relative enhancement ratio.

**Table 3 cancers-14-00827-t003:** Response rates stratified by assessment of EOB-MRI hepatobiliary phase.

Lenvatinib (*n* = 33)	Homogenous Type (*n* = 21)		Heterogenous Type (*n* = 12)	
Response category	mRECIST	RECIST	mRECIST	RECIST
CR/PR/SD/PD	1/7/7/6	0/2/13/6	0/5/4/3	0/1/7/4
ORR	8 (38.1%)	2 (9.5%)	5 (41.7%)	1 (8.3%)
DCR	15 (71.4%)	15 (71.4%)	9 (75.0%)	8 (66.7%)
**Atezolizumab Plus Bevacizumab (*n* = 35)**	**Homogenous type (*n* = 19)**		**Heterogenous type (*n* = 16)**	
Response category	mRECIST	RECIST	mRECIST	RECIST
CR/PR/SD/PD	0/8/9/2	0/4/13/2	0/4/6/6	0/1/9/6
ORR	8 (42.1%)	4 (21.1%)	4 (25.0%)	1 (6.3%)
DCR	17 (89.5%)	17 (89.5%)	10 (62.5%)	10 (62.5%)
**Lenvatinib** **(*n* = 33)**	**Hypointensity type (*n* = 27)**		**Hyperintensity type (*n* = 6)**	
Response category	mRECIST	RECIST	mRECIST	RECIST
CR/PR/SD/PD	1/10/9/7	0/2/17/8	0/2/2/2	0/1/3/2
ORR	11 (40.7%)	2 (7.4%)	2 (33.3%)	1 (16.7%)
DCR	20 (74.1%)	19 (70.4%)	4 (66.7%)	4 (66.7%)
**Atezolizumab Plus Bevacizumab (*n* = 35)**	**Hypointensity type (*n* = 28)**		**Hyperintensity type (*n* = 7)**	
Response category	mRECIST	RECIST	mRECIST	RECIST
CR/PR/SD/PD	0/10/14/4	0/4/20/4	0/2/1/4	0/1/2/4
ORR	10 (35.7%)	4 (14.3%)	2 (28.6%)	1 (14.3%)
DCR	24 (85.7%)	24 (85.7%)	3 (42.9%)	3 (42.9%)

CR, complete response; PR, partial response; SD, stable disease; PD, progressive disease; ORR, objective response rate; DCR, disease control rate; RECIST, Response Evaluation Criteria in Solid Tumors (ver 1.1); mRECIST, modified Response Evaluation Criteria in Solid Tumors.

**Table 4 cancers-14-00827-t004:** Relationship between visual assessment and relative enhancement ratio in EOB-MRI hepatobiliary phase.

		Visual Assessment	
		Homogenous Type (*n* = 40)	Heterogenous Type (*n* = 28)
RER	Hypointensity type (RER < 0.9) (*n* = 55)	39	16
	Hyperintensity type (RER ≥ 0.9) (*n* = 13)	1	12

RER, relative enhancement ratio.

## Data Availability

The data presented in this study are available on request from the corresponding author. The data are not publicly available, as they contain sensitive medical information.

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
