# Peer review of "Evaluating the Role of Hepatobiliary Phase of Gadoxetic Acid-Enhanced Magnetic Resonance Imaging in Predicting Treatment Impact of Lenvatinib and Atezolizumab plus Bevacizumab on Unresectable Hepatocellular Carcinoma"

_cancers, 2022, doi:10.3390/cancers14030827_

Round 1

Reviewer 1 Report

The authors present a study wherein they conclude that hepatobiliary phase of EOB-MRI was useful in predicting the therapeutic efficacy/ outcome of atezolizumab plus bevacizumab therapy on unresectable HCC.

Authors need to describe heterogenous/homogenous subtypes and hyperintense/hypointense subtypes more clearly in the context of HCC and MRI. There is not sufficient information in the introduction.

Authors have recognized and mentioned the limitations of the present study such as small cohort size of patients, limitation of the assays to determine the extent of beta catenin mutations and others.

If possible, the authors must try to address these limitations, at least for the size of the patient cohort, for a conclusive study.

In addition, the authors should also improve on the grammatical errors and typos found in the manuscript.

In addition to

Reviewer 2 Report

Sasaki et al. proposed an interesting work concerning MRI results in HCC management with immunotherapy.

They underlined the role of EOB-MRI in heterogenous lesiosn and its probable impact on management.

Nevertheless, thz study is retrospective, single-center.

I just note some revisions:

Major revision: an external review of the result is necessary, in the aim to shunt a bias. If not, it must be mentioned in discussion.

Minor revisions:

Eventual perspectives were not mentioned.

Analysis taking in account the other risk may probably more interesting.

Chracteristics of population arelacking some data, concerning tolerance for exemple.

Round 2

Reviewer 2 Report

I read with attention revised manuscript and their responses: these responses and the midifications in the text seem to be accurate with my previous comments.